# Transcriptomic and Metabolomic Research on the Germination Process of *Panax ginseng* Overwintering Buds

**DOI:** 10.3390/plants13071041

**Published:** 2024-04-08

**Authors:** Ranqi Li, Yashu Li, Miaomiao Tang, Zhengyi Qu, Cai Shao, Peihe Zheng, Wei Hou

**Affiliations:** 1Institute of Special Animal and Plant Sciences of Chinese Academy of Agricultural Sciences, Changchun 130112, China; 15294651423@163.com (R.L.); lys18231302624@163.com (Y.L.); aatroxmiaomiao@126.com (M.T.); qu9581@163.com (Z.Q.); shaocai2003@163.com (C.S.); 2College of Traditional Chinese Medicine, Jilin Agricultural Science and Technology University, Jilin 132109, China; 3Jilin Key Laboratory of Technological Innovation in the Production and Utilization of Dao-di Herbs, Jilin 132109, China

**Keywords:** ginseng overwintering buds, transcriptomic, metabolomic, plant hormones, arginine, polyamines (PAs)

## Abstract

Ginseng (*Panax ginseng* C. A. Meyer) is a perennial plant with a long dormancy period. While some researchers employ gibberellin and other substances to stimulate premature germination, this method is limited to laboratory settings and cannot be applied to the field cultivation of ginseng. The mechanism underlying the germination of ginseng overwintering buds remains largely unexplored. Understanding the internal changes during the dormancy release process in the overwintering buds would facilitate the discovery of potential genes, metabolites, or regulatory pathways associated with it. In this study, we approximately determined the onset of dormancy release through morphological observations and investigated the process of dormancy release in ginseng overwintering buds using transcriptomic and metabolomic approaches. Our analyses revealed that the germination process of ginseng overwintering buds is regulated by multiple plant hormones, each acting at different times. Among these, abscisic acid (ABA) and gibberellic acid (GA) serve as classical signaling molecules regulating the dormancy process, while other hormones may promote the subsequent growth of overwintering buds. Additionally, metabolic pathways associated with arginine may be involved in the dormancy release process. Polyamines synthesized downstream may promote the growth of overwintering buds after dormancy release and participate in subsequent reproductive growth. This study provides insights into the germination process of ginseng overwintering buds at the molecular level and serves as a reference for further exploration of the detailed mechanism underlying ginseng overwintering germination in the future.

## 1. Introduction

The phenomenon of dormancy is an important adaptative strategy of perennial plants to adverse conditions, and is beneficial to the survival of the species. In 1987, Lang classified the dormancy phenomenon in plants into three types: ecodormancy, paradormancy and endodormancy [1]. In 2007, Rohde used poplar as a model and divided its dormancy process into five stages: I, cessation of cell division; II, establishment of dormancy or loss of responsiveness to growth-promoting signals; III, maintenance of dormancy; IV, release from dormancy state or cell-cycle machinery regaining responsiveness to growth-promoting signals; and V, resumption of cell division [2]. During these five stages, many different levels of material and energy metabolism processes occur in plant cells, and many environmental factors are involved as signals to enter, maintain and release dormancy, such as temperature, light, etc.

Ginseng (*Panax ginseng* C. A. Meyer) is a perennial herb belonging to the genus Panax and the family Araliaceae, and has been used to treat many human diseases, including depression, diabetes, inflammation, senility and tumors [3]. Ginseng seedlings germinate from seeds. From the second year, the stem and leaves of ginseng develop from the overwintering bud, which is located at the top of the root of ginseng, including three scales and internal flower buds [4]. Ginseng has a long growth cycle, typically beginning to blossom and bear fruit in the third year. The growth cycle of ginseng can be divided into five stages: the seedling stage, the leaf unfolding stage, the flowering stage, the fruiting stage and the overwintering dormancy stage. In northeastern China, the dormancy period of ginseng is about seven months, from the end of September to the end of April of the following year. According to Lang’s classification of dormancy, the dormancy type of ginseng is endodormancy. This means that the dormancy process of ginseng is induced by environmental factors (light and temperature) and regulated by the physiological factors of the overwintering buds themselves (such as internal low-temperature response elements, photoperiod response elements, etc.). Even if a suitable environment is provided, the overwintering buds will still not germinate and must go through a certain low-temperature period to be released from hibernation.

Current research on the dormancy regulation mechanism of perennial herbaceous plants focuses mainly on four aspects: the photoperiod [5,6,7], temperature [8,9,10,11], plant hormones [12,13,14,15] and cell cycle [16,17]. Each part is interconnected to form a complex regulatory network. However, until now, there has been no systematic study on the dormancy release process of ginseng overwintering buds. Researchers have only studied the effects of temperature and exogenous plant hormones on the germination of ginseng or American ginseng (*Panax quinquefolius* L.) [18,19,20,21]. These studies have proved that dormant ginseng can germinate in advance through the application of low-temperature treatment (3~5 °C) or exogenous gibberellin. The slow growth rate of ginseng makes it difficult to accumulate medicinally active ingredients quickly. In addition, the dormant period of ginseng accounts for more than half of its complete growth cycle. These factors result in ginseng taking a long time to grow before it can be harvested for medicinal use, at least five years. Shortening the dormancy period may shorten the cultivation cycle and accelerate the cultivation and production of ginseng without affecting other growth stages.

In this study, we conducted morphological observations, at both the macroscopic and microscopic levels, on ginseng overwintering buds during the germination process, delineating the time range of dormancy release. Subsequently, by integrating metabolomics and transcriptomics, we investigated the changes in genes and metabolites during the dormancy release process of ginseng overwintering buds. The purpose was to identify potential genes, metabolites, or regulatory pathways closely associated with the dormancy release process of ginseng overwintering buds, providing insights for elucidating the mechanism of ginseng bud germination in the future.

## 2. Results

### 2.1. Observation on the Appearance of Overwintering Buds and Paraffin Sections of Young Stems

The appearance and cell morphology of ginseng overwintering buds at different stages were observed (Figure 1). In terms of appearance, the overwintering buds of ginseng did not germinate on 16 March 2023 and 30 March 2023. The buds were similar in shape, smooth in surface and the outer scales tightly wrapped the inner flower buds. However, on 6 April 2023, the appearance of ginseng overwintering buds began to change, and by 13 April, the stems inside the overwintering buds had broken the scales. We hypothesized that changes in the cell morphology within the overwintering buds might precede those observed externally. Therefore, we conducted focused observations on consecutive sets of samples using paraffin sections. As anticipated, and judging from the internal cell morphology of ginseng young stems, the parenchyma cells in the center of the young stem were found to be round, or nearly round, before 23 March 2023 and were seen to be closely arranged without an intercellular space. The nucleus was round and located in the middle of each cell. After 30 March 2023, the volume of parenchyma cells gradually increased, the shape of the cells and nuclei became irregular, and the number of nuclei in the visual field gradually decreased. The intercellular space appeared and gradually enlarged. This indicates that, during the period from the 23rd to the 30th of March 2023, the cells inside the ginseng winter buds gradually resumed their life activities at the cellular level from dormancy. However, significant changes in the morphology of the winter buds did not occur until one week later. Therefore, samples collected on 16th March, 23rd March, 30th March, and 6th April adequately represent the four stages of the ginseng winter bud from dormancy to germination (i.e., dormant, pre-germination, cell awakening, and morphological changes). We conducted omics analyses using the corresponding samples, with the four stages labeled as D1, D2, G1, and G2, as illustrated in Figure 1.

### 2.2. Results of Metabolomic Analysis

#### 2.2.1. Quality Evaluation of Metabolome Data

Untargeted metabolomic analysis was conducted on the four sets of samples. The base peak chromatograms revealed similar trends in the curves of the four sample groups (Appendix A), which proved that the LC-MS system worked stably, the data repeatability was good, and the results are reliable. The principal component analysis (PCA) model reflects the overall situation of the data, such as the degree of aggregation or dispersion of samples, so as to preliminarily judge the size of the differences between the data. The results show that the four groups of samples were distributed in different positions on the PCA score map (Figure 2A) and were far away from each other, while the six biological repeats in each group were clustered together, indicating that there were great differences among the samples. Additionally, the repeatability of the samples in the group was good.

Performing a permutation test on the model can determine whether the model has the possibility of overfitting. The results show that all of the Q2 values (blue points) were located lower on the graph than the original Q2 value in the upper right corner (Appendix A). These results further illustrate the reliability of the metabolomic data, which can be used for further analysis.

#### 2.2.2. Screening of Differentially Accumulated Metabolites (DAMs) at Different Stages during the Dormancy Release Process of Ginseng Overwintering Buds

By comparing metabolite information across multiple databases, we identified a total of 7981 metabolites in 4 groups of samples. The results of cluster analysis (Figure 2B) show that the samples in the D1 and D2 stages clustered together, while the samples in the G1 and G2 stages formed another cluster. These findings suggest that significant changes occurred within ginseng overwintering buds between March 23rd and 30th, consistent with the results obtained from paraffin sectioning.

We used the preset *p*-value (*p* < 0.05) and VIP-value (VIP > 1) as thresholds to screen for DAMs between two adjacent groups. Pairwise comparison showed that the number of DAMs between stages D1 and D2, D2 and G1, and between stages G1 and G2 were 209, 223 and 203, respectively. Judging from the changes in metabolite content (Figure 2C), among the three groups of comparisons, the number of up-regulated metabolites between the samples in the D2 and G1 stages was the largest, at 110, while the corresponding values for the other two comparisons were 88 (D1 vs. D2) and 82 (G1 vs. G2). There was little difference in the number of down-regulated metabolites, being 121, 113 and 121, respectively. With the collection of metabolites in the samples in the D1 stage as the control, the changes in metabolites in the samples in the remaining three stages were observed, and a Venn diagram (Figure 2D) was drawn. It can be seen that, with the passage of time, the number of differential metabolites gradually increased. Through a comparison of the samples of the above adjacent groups and an analysis of the changing trend of metabolites, it can be preliminarily concluded that, during the period from 16th March to 13th April 2023, many metabolic activities took place inside the overwintering buds of *P*. *ginseng*, and the intensity gradually increased, reflecting the process of relieving dormancy.

#### 2.2.3. KEGG Pathway Analysis of DAM during Ginseng Overwintering Bud Germination

We performed KEGG pathway analysis on the differential metabolites and enriched the respective 109, 112 and 109 pathways. We used the *p*-value to filter the enriched pathways and used the top 20 pathways in each set of comparison results in order to draw a bubble chart (Figure 3A–C). As shown in this figure, many of the same pathways were significantly enriched in the three sets of comparisons, such as ABC transporters, biosynthesis of plant secondary metabolites, biosynthesis of amino acid, arginine biosynthesis, arginine and proline metabolism and linoleic acid metabolism, etc. This indicates that these pathways continue to play a role in the process of release of the dormancy of ginseng overwintering buds. In addition, we classified the screened differential metabolites (the union of three groups of contrasted differential metabolites) (Figure 3D) in order to analyze the contribution of each type of substance in the dormancy release process of ginseng overwintering buds. The results show that, among the substances that can be identified, amino acids and polypeptides have the largest variety of metabolites, accounting for 11.80%, followed by fatty acids, accounting for 6.37%. It was unclear which category 26.22% of metabolites belong to. The entry named ‘Other’ represented a collection of metabolites that account for no more than 1% of each metabolite.

### 2.3. Result of Transcriptomic Analysis

#### 2.3.1. Overview of Transcriptome Data

Relying solely on metabolome data does not allow one to fully understand the process by which ginseng overwintering buds are released from dormancy. Therefore, we performed transcriptome sequencing on four groups of samples, in order to obtain more conclusions by combining this sequencing with metabolome data. According to the statistics of the original data of each sample, the Q20 and Q30 values of all libraries were greater than 97.5% and 93%, respectively. Clean reads were obtained by filtering the raw data, and, when compared with ginseng reference genes, the proportion of annotated genes in each sample was more than 97%. After that, we conducted a PCA analysis of the data (Figure 4A), the results of which show that the samples of the four stages were separated from each other, while the three biological repeats of each stage were clustered together on the map. This indicates that the sample selection was appropriate, and that the data obtained by sequencing was of high quality, and could be used for subsequent analysis.

In order to better understand the law of gene expression during overwintering bud germination of *P*. *ginseng*, we clustered the transcriptome maps of samples in four stages, and the results are shown in the heat map in Figure 4B. Similarly, the samples in D1 and D2 stages were grouped into one group, and the samples in G1 and G2 stages were clustered into another group. Using |log2fold change| > 1 and *p*-value < 0.05 as the screening threshold, we screened differentially expressed genes (DEGs). The numbers of the differential genes in the three comparisons were 17,379, 25,360 and 9058, respectively (Figure 4C). We found that the number of DEGs identified between G1 and D2 was significantly higher. Therefore, in the process of ginseng overwintering bud germination, the gene expression also changed greatly from 23rd March to 30th March.

#### 2.3.2. Enrichment Analysis of DEGs

The differential genes in the three sets of comparisons were analyzed through gene ontology (GO) enrichment analysis, and 9763, 10,333 and 7680 GO terms were annotated. These terms were classified as cellular components (CC), molecular function (MF), or biological process (BP) (Figure 5A–C). The results show that the terms in the cellular component and molecular function categories compared in the first two groups were highly similar; notable entries in the cellular component category were “plasma membrane,” “cell periphery,” and “cell membrane,” while notable entries in the molecular function category were “signal receptor activity” and “omega-6 fatty acid desaturase activity.” In the biological process category, the first group (D1 vs. D2) had no entries with significantly higher enrichment levels, and the first two relatively higher entries were “protein modification process” and “response to stimulus.” In the biological process category of the second group (D2 vs. G1), “response to stimulus” was significantly enriched, followed by “trehalose metabolism in response to stress” and “response to abiotic stimulus.” As for the results of the third group (G1 vs. G2), in the CC, MF and BP categories, the significantly enriched entries were almost all related to photosynthesis. Within the cellular process category, the top three most enriched terms were all related to thylakoids. Among the molecular function categories, those with higher enrichment levels include “oxidoreductase activity,” “pigment binding,” and “chlorophyll binding.” Among the biological process categories, “photosynthesis, light reactions” was highly enriched. This indicated that the dormancy release process of ginseng overwintering buds had entered a new stage.

Subsequently, Kyoto Encyclopedia of Genes and Genomes (KEGG) enrichment analysis was performed on the differential genes (Figure 5D–F) and the KEGG pathways of each group were screened according to the *p*-value (*p* < 0.05). Bubble charts were also drawn. Similar to the results of GO analysis, many of the enriched pathways in the results of the first two groups were the same, such as those of starch and sugar metabolism, the MAPK signaling pathway, plant hormone signal transduction, and galactose metabolism. These pathways may be related to energy supply and signal transduction during overwintering bud germination. In the third group, pathways related to photosynthesis and circadian rhythm appeared among the significantly enriched results. This was the same as the result of GO analysis, indicating that the overwintering buds at this stage (from 30 March to 6 April) were almost completely recovered and had begun to prepare for the photosynthesis process after germination. In addition, plant signal transduction pathways were active throughout the entire process, indicating that the germination process of ginseng overwintering buds was regulated by plant hormones.

### 2.4. Plant Hormones and Related Genes during Germination

In order to understand the role of phytohormones in the germination of ginseng overwintering buds, we analyzed the phytohormone contents in four groups of samples. We found that the contents of five plant hormones or derivatives changed significantly during the germination process. These were gibberellin A3 (GA_3_), gibberellin A4 (GA_4_), abscisic acid (ABA), 5-hydroxyindoleacetic acid (5-HIAA), and methyl jasmonate (MeJA) (Figure 6A). Among these, the change trend of GA_3_ and GA_4_ was similar, showing a “decline-rise-decline” trend. ABA levels continued to decrease throughout the germination process. The content of 5-HIAA increased significantly in the G1 stage, and then decreased rapidly. The content of MeJA remained at similar levels in the first three groups of samples but increased significantly in the G2 stage.

In order to further explore the relationship between these substances and the germination process, we searched for genes related to these substances from the “phytohormone signal transduction pathway” in the KEGG pathway analysis results of the transcriptome. We identified a total of 19 genes, including 4 gibberellin receptor *GID1* genes, 4 DELLA protein *RGL1-like* genes, 1 PYRABACTIN RESISTANC-LIKE (*PYL*) gene, 4 cytokinin receptor (*CRE1*) genes, 2 myelocytomatosis protein 2 (*MYC2*) genes, 1 auxin response factor (*ARF*) gene, 2 auxin influx carrier (*AUX1*) genes, and 1 transport inhibitor response 1 (*TIR1*) gene. The expression levels of these genes at 4 stages were plotted on a cluster heat map (Figure 6B). The 19 genes were divided into 3 major categories. The first category includes all *GID1* genes, DELLA protein genes and *PYL*. The expression levels of these genes peaked in D2 stage and then gradually declined. The second category includes four *CRE1* genes and one *TIR1*, with high expression levels observed in both G1 and G2 stages. The third category consisted of two *MYC2* genes, two *AUX1* genes, and one *ARF*. The expression levels of these genes showed little variation in the first three stages but increased in G2 stage. By comparing hormone levels with corresponding gene expression, it was found that the expression pattern of *PYL* in the abscisic acid signal transduction pathway and the trend of ABA content were consistent in the latter three groups of samples. *MYC2* emerged as a key factor in the JA signal transduction pathway. The changes in MeJA content in the four stages aligned with the expression trends of the two *MYC2* genes, both of which sharply increased in G2 stage.

### 2.5. Arginine-Related Pathways during Ginseng Overwintering Bud Germination

Comparing the enrichment analysis section in the metabolomics and transcriptomics results, we found that pathways related to arginine were mentioned repeatedly, including “arginine biosynthesis” and “arginine and proline metabolism.” Therefore, we integrated key genes and metabolites in the two arginine-related pathways (Figure 7A). Based on the aforementioned analysis, we consider the week between March 23rd and 30th to be a critical period for the release of dormancy in the overwintering buds. Therefore, we annotated the changes in partial genes and metabolite contents between stages D2 and G1 in the pathway map, with up-regulated genes/metabolites in red and down-regulated genes/metabolites in green. It can be seen in the figure that arginine and ornithine connect the arginine biosynthetic pathway (modules I and II) and the arginine and proline metabolic pathway (module III). According to the separate analysis of each module, in module I, the increase in upstream N-acetylglutamate and the up-regulation of *argE* and *ACY1* gene expression may be the direct reason for the increase in ornithine content. Ornithine may then enter the urea cycle to produce arginine, enter the arginine and proline metabolic pathway to participate in metabolism (module III), and may also produce polyamines (module IV, which belongs to the arginine and proline metabolic pathway, and is listed separately for distinction). Among these products, the contents of trans-4-Hydroxy-D-proline and spermine increased, which may be related to the increase in ornithine. It is worth mentioning that the content of arginine decreased. We speculate that arginine was mainly synthesized from ornithine via the urea cycle, but that arginine was also one of the precursors of the above products. In the early stage of the transition from dormancy to germination, the synthesis of polyamines and other substances in the overwintering buds of *P*. *ginseng* may also consume the existing arginine, while the increased ornithine could not be converted to arginine by the urea pathway in a short timeframe. According to our conjecture, the content of arginine should increase over a period of time. To this end, we constructed a composite diagram in order to illustrate the variations in the content of related substances throughout the germination process (Figure 7B). It can be seen that the contents of these three substances generally show an upward trend during the germination process, and all show obvious changes between the D2 and the G1 stage. As we speculated, the content of arginine decreased in the G1 stage compared with the D2 stage, followed by an increase in G2. These results indicate that arginine is consumed during the initial stage of germination, suggesting that arginine may be one of the key substances required in the dormancy release process of ginseng overwintering buds.

### 2.6. Validation of Transcriptomic Results Using Q-PCR

According to the sequencing results, genes with large expression differences (|log2foldchange| > 1.5), high gene expression (FPKM ≥ 50), and large sequencing depth (read count > 30) were screened for Q-PCR verification among the differential genes. The gene expression levels in the Q-PCR results were compared with the transcriptome results to observe the gene expression changes in the four stages. The expression level of each gene in the D1 stage in the two sets of results was set to 1, and then the expression level in the other three stages of samples relative to the D1 was calculated. The results for each gene were plotted as a composite diagram. It can be seen from Figure 8 that, among the 12 selected genes, only the expression of EVM0022327 was inconsistent between stages D1 and D2, and the expression trend of each other gene was consistent with the results of Q-PCR and the transcriptome. This shows that our transcriptome data are reliable.

## 3. Discussion

### 3.1. The Role of Phytohormones in the Germination of Ginseng Overwintering Buds

Almost all life activities of plants are directly regulated by plant hormones. ABA and GA are currently the most studied pair of hormones related to plant dormancy. ABA plays a key role in the establishment and maintenance of dormancy, which is an important factor in sensing and transmitting signals of changes in the external environment, inducing and maintaining dormancy. However, the function of GA was opposite to that of ABA, promoting the release of dormancy. Changes in the balance of endogenous ABA/GA content can reflect the dormancy process [22]. Studies on a variety of perennial plants have shown that the content of free ABA in winter buds is high during dormancy, but that its content begins to decrease as dormancy is lifted. Moreover, the dormancy release process of winter buds is also accompanied by an increase in gibberellin content [23,24,25]. There is an antagonistic effect between ABA and GA. In a study of the dormancy–germination regulation of cereal seeds, the interaction mechanism between ABA and GA was discussed in detail [26]. RGL2 (RGA-LIKE2) is the key DELLA protein that inhibits germination. GA shows a negative regulatory effect on RGL2 through “GA→GID1→DELLA.” In this study, we found that the expression of the GID1 gene in ginseng overwintering buds reached a peak in the D2 stage, while the expression of the DELLA protein gene declined rapidly in the subsequent two stages (G1 and G2). At the same time, the content of ABA and the expression of the PYL gene also continued to decline. This proved that there was also an interaction network between GA and ABA inside overwintering buds to maintain and release dormancy.

Different hormones have different functions and may work at different times. The genes in different hormone signaling pathways identified in our experiment had different expression patterns. Based on this, the functions and sequence of actions of these plant hormones during germination can be speculated upon. By comparing the gene expression in the GA signaling pathway with the cell morphology observed in the sections, we found no signs of cell division in the D2 stage. However, according to the expression analysis of relevant genes, the GA signaling pathway was already activated at this time. Additionally, compared with the D1 stage, the ABA content in the D2 stage also decreased. Therefore, we further speculate that the dormancy-releasing effect of GA occurred on March 23rd, or even earlier. As the earliest signaling molecule in plants, GA initiates the germination process of overwintering buds. This may explain why the contents of GA_3_ and GA_4_ were highest at the D1 stage. According to previous research on circadian clocks and hormone signaling, it has been speculated that the initiation of the GA signal transduction pathway may be related to the circadian rhythm of the plant itself [27]. This indicates that changes in the external environment may be the fundamental cause of dormancy release. The CRE1 identified in this experiment was highly expressed in the G1 and G2 stages, indicating that cytokinin had begun to work at this time to promote cell division. This corresponded with our paraffin section results, indicating that the cells in the ginseng overwintering buds have initially been released from their dormancy and have begun to enter the growth stage after March 23rd. The auxin-related genes AUX1, TIR1 and ARF were highly expressed mainly in the G2 stages. This shows that the time point at which auxin began to exert its function may have slightly lagged behind that of cytokinin, and that it was mainly involved in the subsequent growth process of ginseng overwintering buds. jasmonic acid is known as a stress hormone and is not generally associated with dormancy. However, there have been studies indicating that this may be related to flower development [28]. In this study, the content of MeJA and the expression of two MYC2 genes increased suddenly in the G2 stage, and the overwintering buds of ginseng contained ginseng flower buds, indicating that jasmonic acid may be involved in the regulation of the development of ginseng flowers after germination.

### 3.2. Arginine and Polyamine Metabolism during the Germination of Ginseng Overwintering Buds

Free amino acids are important nitrogen metabolic intermediates in plants. Amino acid metabolism is closely related to carbohydrate metabolism and various secondary metabolic processes in plants. The content and variation of various amino acids and the regulation of transport in plants are strongly related to the process of plant growth and development [29,30]. As the “nitrogen pool” of plants, the content and proportion of free amino acids change dynamically so as to meet the nitrogen needs of plants at different developmental stages or in different environments [31]. For example, in conifers, arginine forms a large part of the amino acid pool in stored proteins [32]. Durzan used 14C-L-arginine to study the nitrogen metabolism pathway of Picea glauca. during dormancy. By measuring the types and contents of free amino acids in buds and stem tips during dormancy, it was proven that the seasonal variation of amino acids was related to the carbon and nitrogen metabolism of arginine [33]. Arginine plays an important role in nitrogen storage and transport in plants. In vegetative organs, the nitrogen stored in the form of arginine accounts for 90% of the total free nitrogen [34]. Studies have confirmed that arginine exists in a variety of plants as the most important free amino acid during dormancy. The determination of nitrogen-containing compounds in *Curcuma alismatifolia* Gagnep. during dormancy showed that arginine was the main free amino acid present [35]. Low-temperature treatment will speed up the germination of lily bulbs. Research on changes in the composition and content of free amino acids inside lilies during storage has shown that amino acids of the glutamic acid family, such as arginine and glutamic acid, play important roles in bulb metabolism. The most abundant and most varied amino acid was arginine [36]. As a precursor, arginine can be used to synthesize a variety of signaling molecules, such as ornithine, proline, polyamines, etc. In this study, we observed that the content of arginine decreased briefly in the G1 stage, and then increased rapidly; however, the content of some downstream products increased continuously. This proved that arginine is needed during germination in order to produce the substances necessary for plants during this period. Through the analysis of the content of downstream metabolites in related pathways, we speculated that polyamines might be beneficial to the germination of overwintering buds of *P*. *ginseng*.

Polyamines are a class of organic compounds containing two or more amino groups, which are ubiquitous in various organisms, including putrescine, cadaverine, spermidine and spermine. Polyamines play an important role in plant growth and development and in dealing with external adverse environmental stress. Firstly, polyamines may be necessary for cell proliferation. Some studies have shown that the content of several polyamines in cells increases during mitosis, and that the order of increase is consistent with the synthetic order of putrescine, spermidine and spermine. Follow-up work has shown that the content of spermidine is positively correlated with cell growth rate, indicating that the accumulation of spermidine is closely related to cell division [37]. The use of ornithine decarboxylase inhibitors will lead to a decrease in the level of putrescine, which in turn reduces the level of spermidine. At the same time, DNA synthesis and cell proliferation are also inhibited. However, at this time, the application of exogenous polyamines will lead to the recovery of cell proliferation [38]. Studies have shown that the flower bud germination rate of ‘Hosui’ pear (*Pyrus pyrifolia* Nakai) trees has a very significant positive correlation with the content of endogenous spermidine, and a significant positive correlation with the contents of putrescine and spermine [39]. In the process of ginseng overwintering bud germination, the cells gradually regain their activity and begin to proliferate through mitosis; this is a necessary function of polyamines at this stage.

Polyamines have also been proven to be closely related to the reproductive development of many plants. For example, in the reproductive structure of citrus, the content of polyamines accounts for 80% of the total [40]. High levels of spermidine are beneficial to flower formation in tobacco tissue culture [41]. Rey et al. have found that high levels of spermidine and spermine in hazel buds may be related to flowering and growth [42]. Based on the biological characteristics of ginseng, the interior of the ginseng overwintering buds consists of flower buds that have completed differentiation. The ginseng will undergo reproductive growth soon after the overwintering buds germinate. Therefore, we speculated that the activity of the polyamine synthesis pathway may be related not only to vegetative growth during germination, but also to the upcoming reproductive growth process.

In this study, we did not observe significant differences in the contents of putrescine and spermidine. However, the content of spermine increases during the germination process, while putrescine, spermidine and spermine engage in an upstream and downstream relationship in the polyamine synthesis pathway, and the content of precursors such as arginine is limited in the early stages of germination. If putrescine and spermidine are essential substances for germination, it is possible that the content differences in each period will be insignificant, due to their consumption. In short, the continuous increase in spermine content indicates that the polyamine synthesis process is active during the conversion process from dormancy to germination, and that polyamines are essential for the initiation and maintenance of germination. However, with regard to the entire process, the questions of which substances specifically play a role and what their mechanisms of action are require further verification and research.

In addition, the pathway branch in which proline is located was also found to be active in this experiment. Accumulation of the downstream product trans-4-Hydroxy-D-proline was observed in the G1 stage compared with the D2 stage, and the *P4HA* gene in the pathway was significantly up-regulated. High activity of proline biosynthesis may lead to the accumulation of NADP^+^ [43]. This phenomenon most likely activated the pentose phosphate pathway (PPP), which is the major source of NADPH in nonphotosynthetic tissues [44]. Activation of the oxidative pentose phosphate pathway has been thought to be necessary to break bud dormancy. During the pathway screening of the transcriptome results, we found that a large number of genes in energy-related pathways, such as the starch and sucrose metabolism and the galactose metabolism pathway, were significantly up-regulated. Therefore, the branch of proline metabolism may be related to energy metabolism during germination.

Finally, we provide a visual representation of the discussion section (Figure 9), depicting the substances that may be involved at different stages of ginseng germination and its subsequent growth.

## 4. Materials and Methods

### 4.1. Plant Material and Treatment

The four-year-old ginseng roots used in the experiment was sourced from the Medicinal Plant Resources Nursery of the Institute of Special Animal and Plant Sciences of Chinese Academy of Agricultural Sciences (Jilin, China). They were transplanted to the experimental fields behind the research institute’s laboratory building before the ginseng roots entered dormancy (Changchun, China). We planned to roughly estimate the time range of dormancy release by observing the changes in the appearance of overwintering buds, followed by the correction of our results through paraffin sectioning. Therefore, we conducted sampling at regular time intervals. Starting from 1 November 2022, samples were collected every 15 days (sampling dates were 1st and 16th November, 1st and 16th December, and so on). Forty ginseng plants were randomly collected from the experimental field each time, until obvious sprouting phenomena were observed in the collected overwintering buds. However, with the recovery of ginseng’s physiological activities, overly long sampling intervals might result in significant differences between adjacent samples groups, which could affect subsequent experiments. Therefore, starting from 16 March 2023, the sampling interval was adjusted to 7 days. However, not every batch of collected overwintering buds were used in the experiments. We planned to select samples before and after germination through morphological observations. After being excavated from the ground, ginseng plants had their overwintering buds completely trimmed off with a blade. These samples were utilized for paraffin sectioning, nucleic acid extraction, and omics analysis (overwintering buds for paraffin sectioning were peeled off and placed in FAA fixative, while the remaining samples were stored in a −80 °C freezer). Due to the lightweight nature of individual overwintering buds, it was necessary to combine several buds as one biological replicate for omics analysis. Additionally, temperature probe loggers (Crystal Probe, RC-4CH) were installed to record temperature fluctuations near the overwintering buds.

### 4.2. Paraffin Section

We used paraffin sections to observe the microscopic morphological changes of young stems inside the overwintering buds of ginseng, so as to judge the time of dormancy release. The experiment consisted of the following steps. (1) Fixing: the collected ginseng overwintering buds were placed in FAA fixing solution (45% anhydrous ethanol, 6% acetic acid, 5% formaldehyde). As the FAA fixed solution can be used as a preservation solution, subsequent operations were carried out simultaneously after all of the samples were collected. (2) Dehydration: there were five different proportions of dehydrating agents used in the dehydration process of ginseng overwintering buds (Table 1). (3) Wax immersion: after dehydration, instead of taking out the overwintering buds, a small amount of paraffin powder was added to the D2 reagent. The container was placed overnight at 35 °C to make it easier for paraffin to enter the plant tissue in the following steps. Then, the overwintering buds were moved to a mixed solution of paraffin and tert-butyl alcohol (1:1). After 2–4 h, the overwintering buds were transferred to pure paraffin, the paraffin was changed twice, and these buds were soaked in paraffin for 3 h each time. The containers containing mixed solution and pure paraffin were placed in a water bath at 60 °C. (4) Embedding: the wax-impregnated buds were placed in an uncovered cardboard box made from Kraft paper filled with wax to suspend them in the middle of the paraffin. After the paraffin wax had solidified, the carton was stored in a 4 °C refrigerator. (5) Sectioning: the solidified wax blocks covered with material were shaped into trapezoids with a knife and fixed to the base. Continuous strips of wax were cut out by a slicer (Bright, 5040). The thickness of the slices was set to 10 μm. (6) Baking: the wax strips were cut to an appropriate length and fixed to the slide using an adhesive (containing 1 g of gelatin, 2 g of phenol, and 15 mL of glycerol, using distilled water to adjust the volume to 100 mL). Slides with samples from different periods were numbered and arranged on a tarp in a water bath. The water bath was set to 60 °C. After the liquid on the slide was slowly dried, the sections were fixed on the slide. (7) Dewaxing, dyeing and sealing: We used xylene and alcohol with a gradient concentration for dewaxing, and only hematoxylin was used for staining. The slices were sealed with neutral gum.

### 4.3. Non-Targeted Metabolomics Analysis

#### 4.3.1. Samples Preparation

Each set of samples used for detection includes six biological replicates, and each biological replicate consisted of a sample composed of three overwintering buds. These samples were ground into powder with liquid nitrogen and a certain amount of each sample was added into a 2 mL centrifuge tube (200 mg per sample). Then, 600 μL of methanol containing 2-Amino-3-(2-chloro-phenyl)-propionic acid (4 ppm) was added to the centrifuge tube. After oscillating in the vortex mixer (Kylin-bell, Haimen, China) for 30 s, the steel balls were added and the samples were ground at 55 Hz for 60 s using a tissue grinder (Meibi, Zhejiang, China). The samples were treated with ultrasonic cleaner at room temperature for 15 min, followed by centrifugation at 4 °C and 12,000 rpm for 10 min. The supernatant was filtered with a 0.22 μm microporous membrane filter (Jinteng, Tianjin, China), and the filtrate was added to the detection bottle for subsequent LC-MS detection.

#### 4.3.2. Liquid Chromatography Conditions

The LC analysis was performed on a Vanquish UHPLC System (Thermo Fisher Scientific, St. Louis, MO, USA). Chromatography was carried out using an ACQUITY UPLC^®^ HSS T3 (150 × 2.1 mm, 1.8 µm) system (Waters, Milford, MA, USA). The column was maintained at 40 °C. The flow rate and injection volume were set at 0.25 mL/min and 2 μL, respectively. For LC-ESI (+)-MS analysis, the mobile phases consisted of (B2) 0.1% formic acid in acetonitrile (*v/v*) and (A2) 0.1% formic acid in water (*v/v*). Separation was conducted under the following gradient: 0~1 min, 2% B2; 1~9 min, 2~50% B2; 9~12 min, 50~98% B2; 12~13.5 min, 98% B2; 13.5~14 min, 98%~2% B2; 14~20 min, 2% B2. For LC-ESI (−)-MS analysis, the analytes were carried out with (B3) acetonitrile and (A3) ammonium formate (5 mM). Separation was conducted under the following gradient: 0~1 min, 2% B3; 1~9 min, 2~50% B3; 9~12 min, 50~98% B3; 12~13.5 min, 98% B3; 13.5~14 min, 98%~2% B3; 14~17 min, 2% B3.

#### 4.3.3. Mass Spectrum Conditions

Mass spectrometric detection of metabolites was performed on Orbitrap Exploris 120 (Thermo Fisher Scientific, USA) with an ESI ion source. Simultaneous MS1 and MS/MS (Full MS-ddMS2 mode, data-dependent MS/MS) acquisition was used. The parameters were as follows: sheath gas pressure, 30 arb; aux gas flow, 10 arb; spray voltage, 3.50 kV and −2.50 kV for ESI(+) and ESI(−), respectively; capillary temperature, 325 °C; MS1 range, m/z 100–1000; MS1 resolving power, 60,000 FWHM; number of data-dependent scans per cycle, 4; MS/MS resolving power, 15,000 FWHM; normalized collision energy, 30%; dynamic exclusion time, automatic.

#### 4.3.4. Data Analysis

Raw data were converted to mzXML format using the MSConvert tool from the Proteowizard (v3.0.8789). The XCMS (https://www.bioconductor.org/, accessed on 25 May 2023) software package was used to perform peak detection, filtering, and alignment processing to obtain a quantitative list of substances. Public databases such as HMDB (https://hmdb.ca/, accessed on 25 May 2023), massbank (https://massbank.eu/MassBank/, accessed on 25 May 2023), LipidMaps (https://lipidmaps.org/, accessed on 25 May 2023), mzcloud (https://www.mzcloud.org/, accessed on 25 May 2023), KEGG (https://www.genome.jp/kegg/, accessed on 25 May 2023) and the database built by BioNovoGene Co., Ltd. (BioNovoGene, Suzhou, Jiangsu, China) were used for substance identification. In data quality control, substances with RSD > 30% in QC samples were filtered out. The R software (v4.1.0) package Ropls (v 1.34.0) was used to conduct principal component analysis (PCA), partial least squares discriminant analysis (PLS-DA), and orthogonal partial least squares discriminant analysis (OPLS-DA) on the samples data. The data were scaled and plotted with score plots and loading plots to show the differences in metabolite composition between samples. The *p*-value was calculated according to the statistical test. The variable importance on projection (VIP) was calculated via the OPLS-DA dimensionality reduction method, and we calculated the difference between groups using fold change (FC). When the *p*-value was <0.05 and the VIP-value was >1, the metabolite molecule was considered to have statistical significance.

### 4.4. RNA Sequencing

#### 4.4.1. RNA Extraction and Detection

Each set of samples used for detection included six biological replicates, and each biological replicate consisted of a sample composed of three overwintering buds. These samples were ground into powder with liquid nitrogen. Total RNA was extracted from overwintering buds using RNAprep Pure Plant Plus Kit (DP441, Tiangen, Beijing, China). The concentration and purity of total RNA were detected via a microspectrophotometer (Thermo Scientific NanoDrop 2000, St. Louis, MO, USA), and the integrity was detected by means of RNA special agarose gel electrophoresis.

#### 4.4.2. Library Construction and Quality Inspection

mRNA with polyA tail was enriched by the NEBNext Ultra II RNA Library Prep Kit for Illumina (NEB, Ipswich, MA, USA) using Oligo (dT) magnetic beads, and then mRNA was randomly interrupted by bivalent cations. Using fragmented mRNA as the template and random oligonucleotides as primers, cDNA was synthesized. The double-stranded cDNA was purified and repaired at the double end, and then the “A” base was introduced into the 3 ‘end to connect the sequencing connector. cDNA with a length of about 400–500 bp was screened by AMPure XP beads for further PCR amplification. The PCR product was purified by AMPure XP beads again, and finally the library was obtained. Agilent 2100 Bioanalyzer (Agilent, Santa Clara, CA, USA, 2100) and Agilent High Sensitivity DNA Kit (Agilent, 5067–4626) were used for library quality detection. The total concentration of the library was detected using Pico green (Quantifluor-ST fluorometer, Promega, Madison, WI, USA; Quant-iT PicoGreen dsDNA Assay Kit, Invitrogen, St. Louis, MO, USA). Finally, the concentration of the effective library was quantitatively detected by quantitative real-time polymerase chain reaction (qRT-PCR). Finally, using second-generation sequencing technology (Next-Generation Sequencing, NGS) and based on the Illumina sequencing platform, these libraries were sequenced according to the double terminal (Paired-end, PE).

#### 4.4.3. Data Analysis

After sequencing on the computer, the original data in FASTQ format (raw data) were obtained, and the Q20 and Q30 scores were calculated. Raw data were filtered to obtained clean data, and the GC content of the clean data was calculated. The filtered high-quality sequences were compared with the ginseng reference genome (https://ngdc.cncb.ac.cn/gwh/Assembly/27488/show (accessed on 25 May 2023)) using HISAT2 software (v2.2.1). According to the comparison results, the expression of each gene was calculated. We used the adjusted *p*-value and |log2 foldchange| to set the threshold for significant differential expression to filter differentially expressed genes (DEGs). On this basis, the samples were further analyzed via enrichment analysis and cluster analysis.

### 4.5. Q-PCR Verification

Based on the transcriptome sequencing results, significantly differentially expressed genes with large expression differences and high expression levels between samples were selected for Q-PCR verification in order to ensure the accuracy of the sequencing. The total RNA was extracted using the TransZol Up Plus RNA Kit (Transgen Biotech, Beijing, China), and subsequent operations were performed according to the instructions of PerfectStart^®^ Uni RT&Q-PCR Kit (Transgen Biotech, Beijing, China). The reaction conditions were as follows: pre-denaturation at 94 °C for 30 s, 40 cycles of denaturation at 94 °C for 5 s, and annealing and extension at 60 °C for 30 s. The relative expression of genes was calculated using the 2^−ΔΔCt^ method. Each pair of primers was set up with three biological replicates and three technical replicates. We used *PgGAPDH* as the internal reference gene. The primers used to amplify the target genes are listed in Appendix A.

## 5. Conclusions

In this experiment, we observed the appearance and cell morphology of ginseng overwintering buds and determined the time node for the “dormant–germination” transition. The results show that changes at the cellular level preceded morphological changes by about a week. Omics analysis showed that, during the entire process, especially in the first two time periods (16th to 23rd March and 23rd to 30th March), many complex physiological and biochemical changes occurred in the overwintering buds, including the synthesis of multiple substances, metabolic processes, and signal transduction processes, and these increasingly enhanced life activities mark the recovery of overwintering buds from dormancy. Based on the transcriptomic results, we also found that, during the period from 30th March to 6th April, a large number of photosynthesis-related genes began to be expressed in ginseng overwintering buds. However, at this time, the overwintering buds were still buried in the soil (the overwintering buds were about 15 cm away from the ground surface) and had not yet been stimulated by light. Therefore, these expressed genes prepare the seedlings for photosynthesis after they reach the ground. In addition, we noticed that, during the period from March 16th to March 30th, the pathways with higher enrichment in the transcriptomic GO analysis results were “response to stimulus” and “response to abiotic stimulus.” Combined with the results of our temperature monitoring near overwintering buds (Appendix A, the two data labels on the curve correspond to 16th March and 30th March), we speculated that the “stimulus” or “abiotic stimulus” here may refer to a sudden increase in temperature, because, as mentioned before, the germination process is stimulated by subzero temperatures, which may act as an environmental signal to activate or promote overwintering buds.

In addition, through our research, we have proven that there is a regulatory network composed of multiple hormones during the germination process of ginseng overwintering buds. Among these, ABA and GA play important roles in the maintenance of dormancy and the initiation of dormancy. In addition, some hormones had not previously been identified in the differential metabolites, though their effects could be inferred from the expression of related genes. Among these, cytokinin and auxin substances promote cell division after the dormancy release signal led by GA occurs, meaning that overwintering buds can completely recover from dormancy to the growth state. MeJA may be related to flower development after germination and subsequent reproductive growth. In addition, metabolic pathways centered on arginine were significantly mobilized, and these pathways play an important role in promoting germination. It would be helpful for us to explain the mechanism of ginseng overwintering bud germination in the future. However, the molecular mechanisms within each branch and how they coordinate with each other require further study. In addition, polyamine products may be necessary for the germination process, though it was unclear which product was the specific product that plays the dominant role and this requires follow-up experiments to verify. In the future, we may be able to use this to try to develop new dormancy-releasing substances for use in experiments.

## Figures and Tables

**Figure 1 plants-13-01041-f001:**
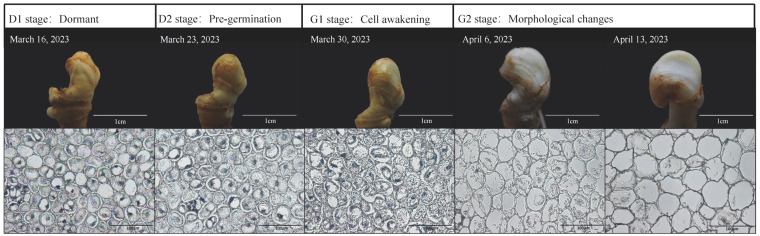
The appearance of ginseng overwintering buds and cross-sectional microsections of young stems. The sampling dates are labeled in the top left corner of each overwintering bud photograph. The top information indicates the different stages of germination process.

**Figure 2 plants-13-01041-f002:**
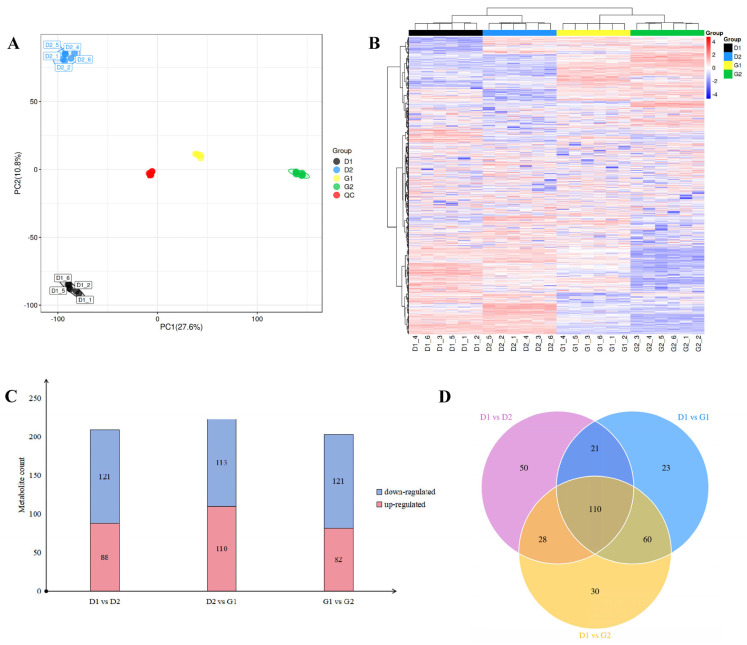
(**A**) PCA score plot of the four groups of samples. The black, blue, yellow, and green dots represent the samples in different stages (D1, D2, G1, and G2, respectively). Red represents the quality control (QC) samples. (**B**) Overall metabolite clustering heatmap. (**C**) Distribution of differential metabolites under pairwise comparison in an up- and down-regulated metabolite quantity histogram. (**D**) Venn diagram reflecting the number of differential metabolites over time.

**Figure 3 plants-13-01041-f003:**
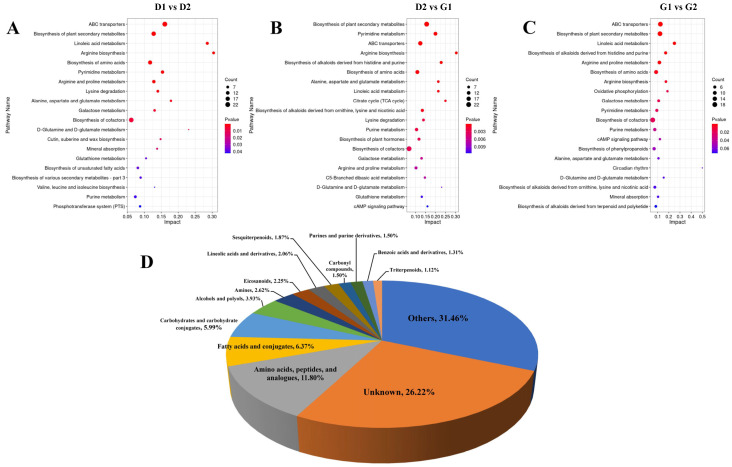
Analysis of differential metabolites during the germination of ginseng overwintering buds. (**A**–**C**) KEGG pathway bubble plots of differential metabolites between samples. (**D**) Categorized pie chart of all of the differential metabolites.

**Figure 4 plants-13-01041-f004:**
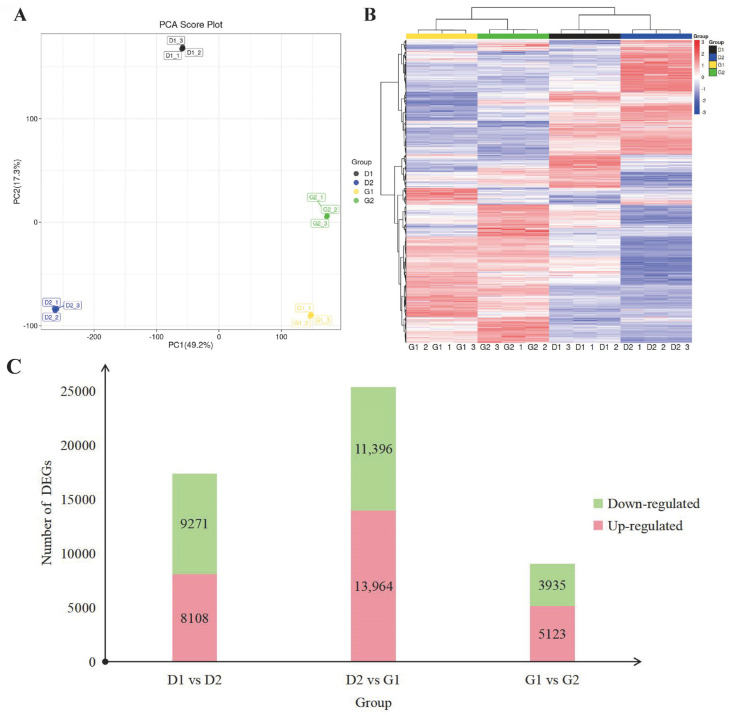
(**A**) PCA score plot of samples. (**B**) Clustering heat map of DEGs. (**C**) Distribution of DEGs under pairwise comparison, shown in an up- and down-regulated genes number histogram.

**Figure 5 plants-13-01041-f005:**
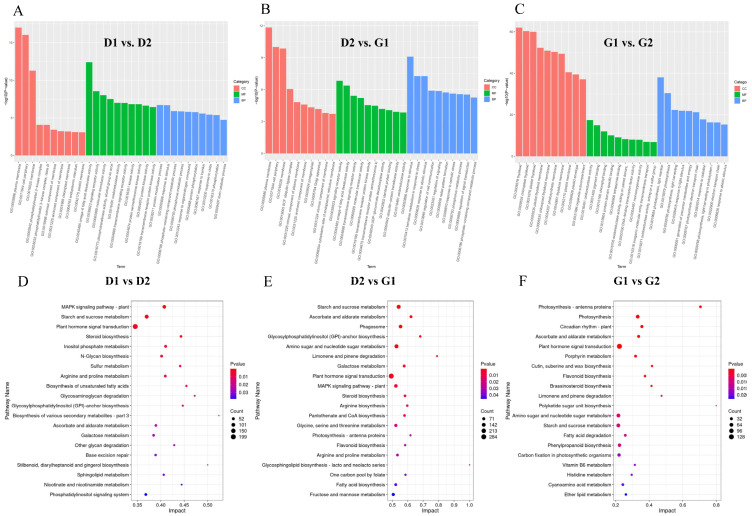
Enrichment analysis results of differential genes between samples. (**A**–**C**) GO enrichment analysis results. (**D**–**F**) KEGG enrichment analysis results.

**Figure 6 plants-13-01041-f006:**
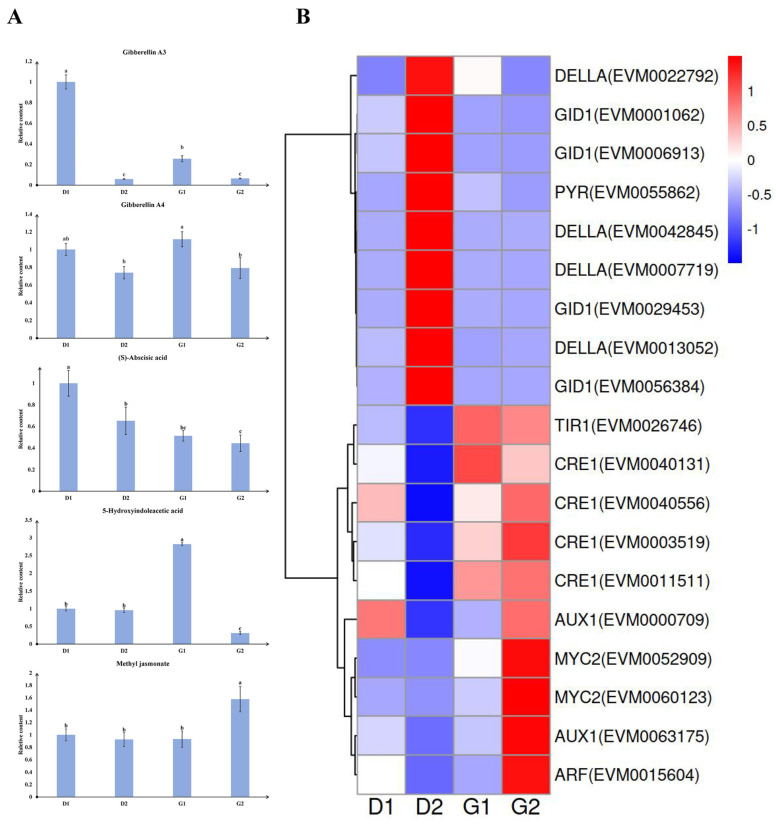
Changes in hormone content and related gene expression during germination. (**A**) Histogram of relative level changes of five hormones. The content of each substance in the D1 stage was normalized to 1, and the contents in other three stages were relative values. Lowercase letters indicate significant differences between groups (*p* < 0.05). (**B**) Clustering heat map, representing the expression of hormone-related genes in four stages.

**Figure 7 plants-13-01041-f007:**
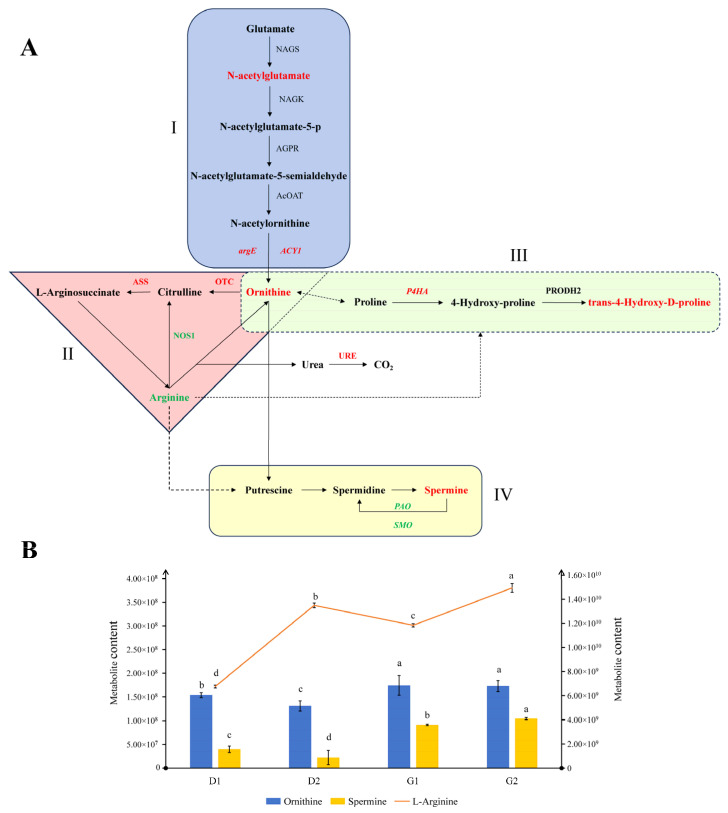
(**A**) Major parts involved in the arginine biosynthetic pathway and proline and arginine metabolic pathways. We divided these into four modules: I. ornithine synthesis; II. urea cycle (arginine synthesis); III. proline metabolism; IV. polyamine synthesis. Up-regulated genes/metabolites are marked in red font, and down-regulated genes/metabolites are marked in green font. (**B**) Changes in the contents of ornithine, arginine and polyamines in the four stages. The main Y-axis (**left**) corresponds with the bar chart, and the secondary Y-axis (**right**) corresponds with the line chart. Lowercase letters indicate significant differences between groups (*p* < 0.05).

**Figure 8 plants-13-01041-f008:**
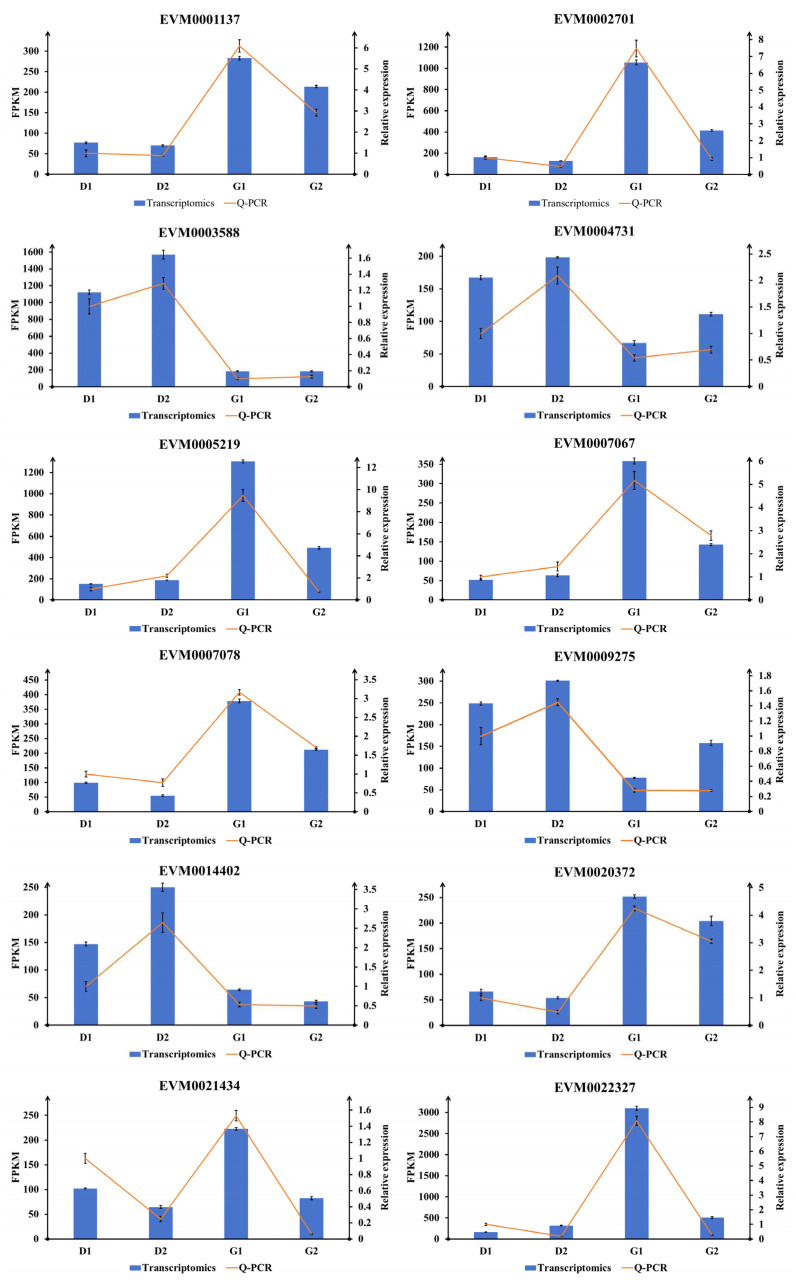
Q-PCR validation of 12 genes. The histograms represent the FPKM value of each gene in the transcriptome results, and the broken lines represent the relative expression of each gene in different samples in the Q-PCR results. In the Q-PCR results, the expression level of each gene in the D1 stage was set to 1, and the expression level in other stages was the relative expression level relative to D1.

**Figure 9 plants-13-01041-f009:**
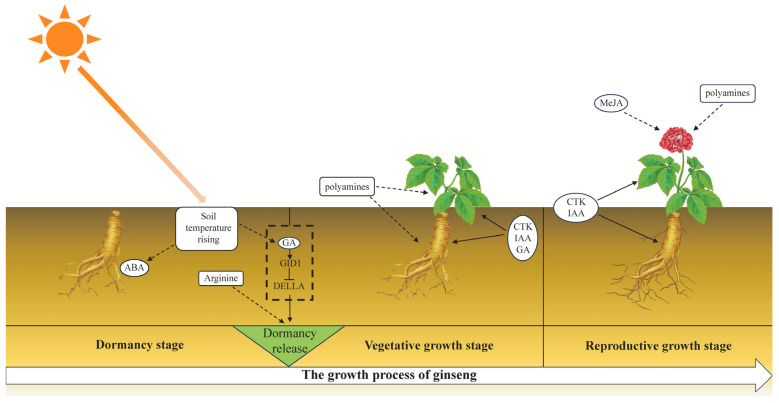
Substances regulating various growth stages of ginseng. Solid lines with arrows indicate regulatory relationships between them. Dashed lines with arrows indicate potential direct or indirect regulatory pathways between them. Dashed boxes represent the regulatory mechanisms of GA promoting plant germination that have been elucidated, which may also operate in ginseng.

**Table 1 plants-13-01041-t001:** The composition ratio of dehydrating agent and the processing time of each step.

Reagent Proportion and Code Name	A	B	C	D_1_	D_2_
Distilled water:70% ethanol:TBA	3:5:2	2:2:4	0:2.5:7.5	TBA	TBA
Dehydration time	1–2 h	1–2 h	1–2 h	1 h	1 h

Note: TBA is TERT-butyl alcohol.

## Data Availability

The data presented in this study are available on request from the corresponding author.

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
