# Peer review of "Transcriptomic and Metabolomic Research on the Germination Process of Panax ginseng Overwintering Buds"

_plants, 2024, doi:10.3390/plants13071041_

Round 1

Reviewer 1 Report

Comments and Suggestions for Authors

-       This study examines the process of dormancy release in ginseng overwintering buds through transcriptomic and metabolomic approaches, revealing the involvement of multiple plant hormones and metabolic pathways. The study provides insight into the molecular-level germination process of ginseng overwintering buds, serving as a reference for further exploration of the detailed mechanism underlying ginseng overwintering germination. The text of the paper is factual, concrete, realistic, and understandable. However, there are important flaws in the manuscript which are listed below:

- In the entire manuscript, the scientific name "Panax ginseng" must only be mentioned in its full form for the first time. For subsequent mentions, it can be abbreviated as "P. ginseng". Additionally, all scientific names of species mentioned in references should be italicized.

- In the section on plant materials and treatment line 515, the authors mention that several buds were combined as one biological replicate for omics analysis. However, it remains unclear why only one biological replicate was used. Perhaps they could have divided the samples for at least three biological replicates.

- The involvement of multiple plant hormones and metabolic pathways in the process suggests a highly intricate and interconnected regulatory network, which may present challenges in fully understanding the mechanisms. It is recommended that this point be discussed in the manuscript.

Comments on the Quality of English Language

Minor editing of English language required

Reviewer 2 Report

Comments and Suggestions for Authors

Interesting paper of good quality with sufficient data. Authors are encouraged to revise their introduction, avoid statements without references, provide some more details in the methodology section and justify their choices and approaches.

Furthermore, some specific comments can be found below:

l.34-40: The phenomenon of dormancy is an important adaptative strategy of perennial plants to adverse conditions, which is beneficial to the survival of the species. In 1987, a simplified definition of dormancy—as a temporary suspension of visible growth of any plant structure containing a meristem was given by Lang, and he divided dormancy  into three groups: Ecodormancy, Paradormancy and Endodormancy [1]: It does not add anything in thestudy, as authors say this is a simplistic definition. I would omit that.

l.56-58: The dormancy period is about seven months, from the end of September to the end of April of the following year: this is valid regarddless of where the crop is grown? If not, the delete this.

l.59-64: This means that the dormancy process of ginseng is induced by  environmental factors (light and temperature) and regulated by the physiological factors of the overwintering buds themselves (such as internal low-temperature response 61 elements, photoperiod response elements, etc.). Even if a suitable environment is provided, the overwintering buds will still not germinate and must go through a certain low-temperature period to be released from hibernation: some references are missing here

l.81: level instead of levels

l.94: signs of germination is totally non-scientific term at all. Be specific!

l.274: As can be seen...? Revision is necessary

In Fig.8 standard errors seem surprisingly low: anyexplanation on that?

l.495: Plant material instead of Plant materials

l.496-500: some more info on the origin of the tested plant material is required

l.502: 40 instead of forty

l.509: Avoid future statements (eg we will select), the study has been already conducted

Comments on the Quality of English Language

Moderate English quality, some suggestions are given above
